# Safety of Inhomogeneous Dose Distribution IMRT for High-Grade Glioma Reirradiation: A Prospective Phase I/II Trial (GLIORAD TRIAL)

**DOI:** 10.3390/cancers14194604

**Published:** 2022-09-22

**Authors:** Patrizia Ciammella, Salvatore Cozzi, Andrea Botti, Lucia Giaccherini, Roberto Sghedoni, Matteo Orlandi, Manuela Napoli, Rosario Pascarella, Anna Pisanello, Marco Russo, Francesco Cavallieri, Maria Paola Ruggieri, Silvio Cavuto, Luisa Savoldi, Cinzia Iotti, Mauro Iori

**Affiliations:** 1Radiation Oncology Unit, Azienda USL-IRCCS di Reggio Emilia, 42123 Reggio Emilia, Italy; 2Medical Physics Unit, Azienda USL-IRCCS di Reggio Emilia, 42123 Reggio Emilia, Italy; 3Neuroradiology Unit, Azienda USL-IRCCS di Reggio Emilia, 42123 Reggio Emilia, Italy; 4Neurology Unit, Neuromotor and Rehabilitation Department, Azienda USL-IRCCS di Reggio Emilia, 42123 Reggio Emilia, Italy; 5Clinical Trials and Statistics Unit, Azienda USL-IRCCS di Reggio Emilia, 42123 Reggio Emilia, Italy

**Keywords:** glioblastoma multiforme, GBM, re-irradiation, recurrence, recurrent GBM, voxel-based dose escalation, brain retreatment

## Abstract

**Simple Summary:**

Glioblastoma multiforme (GBM) is the most frequent primary malignant brain tumor, and despite advances in imaging techniques and treatment options, the outcome remains poor and recurrence is inevitable. Salvage therapy presents a challenge, and re-irradiation can be a therapeutic option in recurrent GBM. The decision-making process for re-irradiation is a challenge for radiation oncologists due to the expected toxicity of a second course of radiotherapy and the limited radiation tolerance of normal tissue; nevertheless, it is being increasingly used, as several studies have demonstrated its feasibility. The current study aimed to investigate the safety of moderate–high-voxel-based dose escalation radiotherapy in recurrent GBM patients after conventional concurrent chemoradiation. Twelve patients were enrolled in this prospective single-center study. Retreatment consisted of re-irradiation with a total dose range of 30–50 Gy over 5 days using the IMRT (arc VMAT) technique using dose painting planning. The treatment was well tolerated. No toxicities greater than 3 were recorded; only one patient had severe G3 acute toxicity, characterized by muscle weakness and fatigue. Median overall survival (OS2) and progression-free survival (PFS2) from the time of re-irradiation were 10.4 months and 5.7 months, respectively. Our phase I study demonstrated an acceptable tolerance profile of this approach, and the future prospective phase II study will analyze the efficacy in terms of PFS and OS.

**Abstract:**

Glioblastoma multiforme (GBM) is the most aggressive astrocytic primary brain tumor, and concurrent temozolomide (TMZ) and radiotherapy (RT) followed by maintenance of adjuvant TMZ is the current standard of care. Despite advances in imaging techniques and multi-modal treatment options, the median overall survival (OS) remains poor. As an alternative to surgery, re-irradiation (re-RT) can be a therapeutic option in recurrent GBM. Re-irradiation for brain tumors is increasingly used today, and several studies have demonstrated its feasibility. Besides differing techniques, the published data include a wide range of doses, emphasizing that no standard approach exists. The current study aimed to investigate the safety of moderate–high-voxel-based dose escalation in recurrent GBM. From 2016 to 2019, 12 patients met the inclusion criteria and were enrolled in this prospective single-center study. Retreatment consisted of re-irradiation with a total dose of 30 Gy (up to 50 Gy) over 5 days using the IMRT (arc VMAT) technique. A dose painting by numbers (DPBN)/dose escalation plan were performed, and a continuous relation between the voxel intensity of the functional image set and the risk of recurrence in that voxel were used to define target and dose distribution. Re-irradiation was well tolerated in all treated patients. No toxicities greater than G3 were recorded; only one patient had severe G3 acute toxicity, characterized by muscle weakness and fatigue. Median overall survival (OS2) and progression-free survival (PFS2) from the time of re-irradiation were 10.4 months and 5.7 months, respectively; 3-, 6-, and 12-month OS2 were 92%, 75%, and 42%, respectively; and 3-, 6-, and 12-month PFS2 were 83%, 42%, and 8%, respectively. Our work demonstrated a tolerable tolerance profile of this approach, and the future prospective phase II study will analyze the efficacy in terms of PFS and OS.

## 1. Introduction

Glioblastoma multiforme (GBM) is the most aggressive astrocytic primary brain tumor, classified as a grade IV tumor according to the WHO, which arises in 15–20% of all primary intracranial tumors [1]. Concurrent temozolomide (TMZ) and radiotherapy (RT) followed by maintenance of TMZ is the current standard of care [2]. Despite advances in imaging techniques and multi-modal treatment options, the median overall survival (OS) remains poor, ranging from 12 to 18 months [3,4]. Recurrence is inevitable, and its management is often case dependent. Several studies have shown that recurrent GBM (rGBM) most often occurs within 2–3 cm from the border of the original lesion [5,6,7,8]. Multiple lesions, which do not show patterns of continuous growth, intraventricular spread, or dissemination, occur in less than 5% of patients [9]; however, metastases are extremely rare [10]. 

The first challenge for the clinician is the differential diagnosis between disease recurrence and treatment-related complications such as pseudoprogression/radionecrosis, which occurs in 20–30% of cases [11,12,13,14]. Usually, pseudoprogression (PP) is characterized by tumor volume increase within 3 months post-chemoradiation therapy, but delayed cases have been reported [15,16], while radionecrosis (RN) appears 3–12 months after RT [17,18,19,20,21,22]. Several studies have investigated how to distinguish rGBM from RN, but the best diagnostic method is not yet well defined [23,24,25,26,27,28,29,30,31,32,33]. Concerning treatment of rGBM, less than 50% of these patients are eligible for second surgery (12–48%) [34,35,36]; however, when feasible, it is associated with increased OS (i.e., 5–11 months) and a preservation of the neurological status [31,32,33,34,35]. As an alternative to surgery, re-irradiation (re-RT) can be a therapeutic option in selected cases. As for other oncological pathologies, the decision-making process of re-irradiation is a challenge for radiation oncologists due to the expected toxicity and the limited radiation tolerance of normal tissue [36,37,38,39].

Re-RT for brain tumors is increasingly used today, and several studies have demonstrated its feasibility. Besides differing techniques, the published data include a wide range of doses, emphasizing that no standard approach exists. The comparison between these studies is extremely complex, but the median overall survival reported is between 7 and 19 months. A secondary analysis by the Radiation Therapy Oncology Group (RTOG) 0525 trial demonstrated a modest clinical benefit of re-RT compared to best supportive care alone that increases when re-RT is combined with systemic therapies [40,41]. A systematic review and a metanalysis of 50 studies support the benefit of re-RT with a 6-month PFS of 43% [42]. However, the lack of prospective trials has limited the ability to draw robust conclusions in rGBM treatment. Using modern precision RT techniques (stereotactic radiosurgery (SRS) [43], stereotactic radiotherapy (SRT), or intensity-modulated radiation therapy (IMRT), re-irradiation has proven to be a feasible and safe option treatment with possible benefits in terms of outcome and to minimize dose to organs at risk (OAR) previously treated in the field. In particular, stereotactic radiosurgery has already shown good results with a better 6-month PFS (47%) [42]. The main risk of GBM re-irradiation is RN. Mayer et al., analyzing 21 re-irradiation studies, reported that a major contribution to RN was the total dose received and that there was no correlation between time to re-irradiation or total cumulative dose greater than 100 Gy [44]. More recently, Scoccianti et al. published a comprehensive review of the literature in order to provide practical suggestions for daily clinical practice, stressing that re-irradiation must be tailored to the individual patient by balancing the risk of toxicity with the eventual benefit [45]. A variety of RT regimens has been used, and therefore no consensus on optimal total dose and fractionation scheme is available. The rationale for hypofractionation is based on increased tumor cell kills as a direct result on DNA double-strand break, as well as reduction in tumor cell repopulation. On the basis of these data, a prospective experimental trial on dose-escalated stereotactic radiotherapy using an MRI voxel-based dose painting technique in rGBM was carried out at our institute. The premise for our trial was based mainly on the following aspects: (1) GBM is considered a radioresistant malignancy, and the majority of patients had local recurrence within the field of re-irradiation [46]; (2) modern radiation techniques could potentially allow dose escalation within smaller high-risk regions without exceeding the tolerance dose of the surrounding brain tissue.

The current study aimed to investigate the safety of moderate–high-voxel-based dose escalation in recurrent GBM patients after conventional concurrent chemoradiation.

## 2. Materials and Methods 

### 2.1. Inclusion Criteria

All participants had to meet the following inclusion criteria: (1) age of more than 18 years old; (2) Karnofsky Performance Status (KPS) ≥ 70 or ECOG Performance Status 0–2; (3) estimated survival of at least 3 months; (4) relapsed GBM after standard surgery radio-chemotherapy treatment; (5) secondary GBM having progressed from low-grade diffuse astrocytoma or anaplastic astrocytoma (AA) or anaplastic oligodendroglioma (AO) previously treated with RT and TMZ; (6) unequivocal evidence of tumor progression by MRI; (7) previous RT with therapeutic doses (45 to 60 Gy at 2 to 3 Gy per fraction); (8) at least 3 months from the end of first RT course.

### 2.2. Exclusion Criteria

Exclusion criteria were as follows: (1) newly diagnosed with GBM; (2) KPS < 70%, ECOG > 3; (3) pregnant women or nursing mothers; (4) any concurrent medical or psychological condition that could compromise the ability to provide informed consent; (5) patient unable to follow procedures or follow-up visits.

### 2.3. Study Design

Assuming that the dose painting could increase the chance of cure at minimized radiation-induced toxicity in stereotactic radiation therapy (SRT) for recurrent GBM, we conducted an exploratory, prospective, experimental, single-center study, with the primary goal of testing the safety of this approach. After identifying the main safety endpoint (early toxicity) and in the absence of data available from previous studies that would be useful for sizing the sample, the sample was defined, according to opportunity and feasibility criteria, as 12 patients. Indeed, the primary endpoint was the safety, i.e., assessment of acute toxicity reported within 3 months of the end of reirradiation.

This protocol was divided into three steps:–Step 1 included a sequential treatment and evaluation of the first 3 subjects; –Step 2 included a sequential treatment and evaluation of 3 other patients;–Step 3 included a sequential treatment and evaluation of the last 6 patients.

At the end of each step, safety monitoring was performed, and a report was produced and evaluated by an external experts advisory board. All subjects in step 2 and step 3 were allowed to continue enrollment in the study only after safety monitoring and if the treatment was well tolerated and no toxicity above G3 was evident in the previously treated patients.

### 2.4. Imaging

The patients underwent a computed tomography (CT) scan on a GE HiSpeed NX/I, and 2 mm of strata reconstruction was set. Subsequently, all patients underwent an MRI scan on a Philips Achieva 1.5 T scanner with an 8-channel phased-array head coil. The examination included a 3D T1-weighted, a 3D fluid attenuated inversion recovery (FLAIR) and a diffusion-weighted sequence. An apparent diffusion coefficient (ADC) map was calculated pixel-by-pixel with software on the MR scanner console according to the following formula:ADC = 1/(b − b0) log (S0/Sb)
where b0 and b values were set to 0 and 1000 s/mm^2^, respectively. S0 and Sb are the pixel values in the images with b0 and b, respectively.

The CT and MRI datasets were exported to the oncology information system Aria (Varian, Palo Alto, USA), and a rigid registration was automatically performed with the “image registration” tool. A radiation oncologist checked the accuracy of the registration process and manually corrected the registration if needed. Co-registered CTs and MRIs were used to define the target volume and organs at risk (OARs).

Target volume delineation was performed using the ARIA^®^ Oncology Treatment Planning System (TPS). The gross tumor volume (GTV) corresponded to the contrast enhancement visible lesion on MRI. Clinical target volume (CTV) was not created; therefore, the two volumes GTV and CTV coincided in our series. An isotropic 5 mm expansion was applied to the GTV/CTV to obtain the planning target volume (PTV). The outlined OARs were: brainstraim, lens, eyes, optical chiasma, optical nerves, and normal brain tissue.

### 2.5. Planning

In order to perform the dose painting by numbers (DPBN) planning task, the same workflow described by Orlandi et al. [47] was used (Figure 1). In line with the literature evidence, we related the doses to the ADC values and prescribed the highest value of the dose to the lowest ADC values (higher cellularity and higher number of clonogens).

In DPBN, a continuous relation between the voxel intensity of the functional image set and the risk of recurrence in that voxel were assumed [48,49]. A minimum dose of 30 Gy (Dmin) in five consecutive fractions to the planning target volume (PTV) with the maximum gross tumor volume (GTV) dose escalated up to 50 Gy were prescribed. Prescription isodose lines were chosen to at least encompass 95% of the PTV with the minimum prescription dose, with no more than 1 cc of PTV receiving doses >50 Gy and no more than 1% of the PTV receiving <95% of the prescribed dose. The protocol required that the cumulative brainstem dose not exceed 70 Gy at the equivalent dose of 2 Gy per fraction; the remaining normal tissues should be constrained as much as possible without compromising the target volume coverage. An example of T1, ADC MRI maps, and dose distribution are shown in Figure 2.

To convert imaging data into prescription dose, a linear function was used, as described by Bowen et al. [50]. The conversion function is defined in Equation (1).
(1)$$D_{i}\left(ADC_{i}\right)=\left\{\begin{array}{cc} & D_{max},\;ADC_{i}\leq ADC_{LO}\\ D_{min}+\frac{ADC_{HI}-ADC_{i}}{ADC_{HI}-ADC_{LO}}\left(D_{max}-D_{min}\right), & ADC_{LO}<ADC_{i}<ADC_{HI}\\ & D_{min},\;ADC_{i}\geq ADC_{HI} \end{array}\right.$$
where *D_i_* and *ADC_i_* represent the dose and the *ADC* values of the *i*-th voxel of the gross tumor volume (GTV), respectively; ADCLO and ADCHI are the 90th percentile and 10th percentile of ADC value distribution, respectively.

To transform the DPBN dose prescription maps into structures and to proceed with plan optimization on the structure-based TPS (Eclipse V13.7, Varian, Palo Alto, CA, USA), a finite number of dose levels LK (1 ⩽ K ⩽ 9) was chosen using patient-dependent criteria. Therefore, the specific dose value LK of that level was assigned for each *i*-th voxel, pertaining to the LK levels. The set of LK that best fitted the patient-specific DPBN prescription map was obtained in order to minimize the overall GTV quality factor (QF) [51]. To generate DPBN dose prescription maps and to find the best LK sets, a home-made API script and MATLAB script were used, respectively. All plans were performed using the VMAT technique, and several coplanar and non-coplanar arcs were set on the basis of the PTV and organ setup of each case. Six MeV flattening filter-free fields were used. Before treatment, according to the specific internal protocol, patient-specific quality assurance controls (QA) were performed for each plan.

### 2.6. Treatment

Stereotactic RT was administered in 5 consecutive daily sessions within 5 days of performing MRI. The prescription dose within the PTV ranged from a minimal dose of 30 Gy (6 Gy per session) for the areas with the highest apparent diffusion coefficient (ADC) to a gradual dose increase in the areas with lower ADC (establishing 1 cc the maximum volume that receives a dose greater than 50 Gy). Daily image guidance with cone-beam CT was performed before each fraction. A quality assurance protocol was performed: patient charts and treatment plans were reviewed by a panel consisting of investigators and internal–external experts.

### 2.7. Follow-Up

The assessment of radiological and clinical response was based on T1w, T2w, FLAIR, and DWI MRI sequences. We also used dynamic susceptibility contrast-enhanced (DSC) MRI to establish progression versus pseudoprogression or radiation-induced radionecrosis. In doubtful cases, we also requested choline-PET/CT to differentiate progressive disease (PD) or treatment-related changes. The assessment of toxicity was based on neurological clinical examination. Patients were followed up until disease progression or death. All toxicities were recorded and graded according to the NCI CTCAE (National Cancer Institute Common Toxicity Criteria for Adverse Events, version 4.0 [52]). Recurrences were defined as “in-field” if the 95% isodose surface contained >80% of the tumor recurrence, and as “marginal” if they contained 20–80% of the recurrence volume. In all other cases, recurrences were considered outside the radiation field [46]. Regarding the evaluation of quality of life (QoL), the EORTC QLQ-C30 questionnaire version 3.0 was used [46,53]. Each patient completed the QoL questionnaires at four different time points: the start of RT (T1); the end of RT (T2); 1 month after completion of RT (T3); 6 months after completion of RT (T4).

### 2.8. Statistical Analysis

The overall survival (OS2) and progression-free survival (PFS2) functions, calculated from the time of re-irradiation, were estimated using the Kaplan–Meier technique. Thus, the related median time calculations were based on these estimates. Due to the very small sample size, we believe that the range is more informative than the standard error of the median, so we only provide the former.

Scores related to the responses across all examined health-related domains (EORTC QLQ-C30) for the “General quality of life” and “Functioning in social role” were compared across time using the Friedman test with post hoc exact all-pair comparisons tests, according to Eisinga; Bonferroni adjustment was applied to correct for multiple testing. Box and whiskers plots were also made to facilitate the interpretation of the findings. A *p* ≤ 0.05 was considered to indicate statistical significance. Statistical analysis regarded aforementioned scores was performed using R statistical software version 4.0.5. (R Foundation for Statistical Computing, Vienna, Austria).

### 2.9. Ethics Approval and Consent to Participate

The present study was approved by the Ethics Committee of the IRCCS-AUSL Reggio Emilia (approval number GLIORAD 15-18 approved on 16 September 2015, ClinicalTrials.gov identifier number: NCT04610229) and carried out in accordance with the ethical standards laid down in the 1964 Declaration of Helsinki. In accordance with Italian legislation, written informed consent was obtained from all patients.

## 3. Results

From 2016 to 2019, 12 patients met the inclusion criteria. Baseline patient demographics and tumor characteristics are summarized in Table 1.

All patients received a full course of RT with a total dose ranging between 45 Gy in 15 fractions and 60 Gy in 30 fractions, concurrently with TMZ plus adjuvant TMZ. All relapses were localized within the previous high-dose radiotherapy (RT) fields. The median time between primary RT and re-irradiation was 11.0 months (range: 5.6–34.9 months). All patients received corticosteroids at the time of re-irradiation and continued it throughout the radiation treatment until at least the first check-up after 1 month. 

### 3.1. Treatment Characteristics

Median gross recurrence tumor volume was 22.95 cm^3^ (range: 5–69.6 cm^3^). Median re-treatment PTV was 45.7 cc (range: 15–97.13 cm^3^). No dose constraint violations were registered. All patients completed the prescribed SRT; none stopped or delayed treatment. 

### 3.2. Outcomes and Toxicities

Re-irradiation was well tolerated in all treated patients. Toxicities and outcomes are summarized in Table 2. 

No toxicities greater than G3 were recorded; one patient (number 2) had severe G3 acute toxicity, characterized by muscle weakness and fatigue. In the first 3 months, all patients experienced acute G1-G2 toxicity; the most frequently reported symptoms were headache, muscle weakness, fatigue, moderate cognitive disorders, and irritability. At the sixth month follow-up of the seven surviving patients, three had resolved the related toxicity symptoms thanks to steroid therapy, while the others continued to show G1 or 2 toxicity. At 1 year after treatment, a slight toxicity remained. 

Regarding KPS, half of the patients did not deteriorate during the first 3 months after re-irradiation. At the first MRI examination, no complete response (CR) or partial response (PR) occurred. Stable disease (SD) was achieved in four patients (33.3%), with an overall objective response rate of 30%. Progressive disease was observed in four patients, with a PD rate of 33%. Pseudoprogression (PP) occurred in only one patient, with a PP rate in our study of 8%. The remaining three patients are not evaluable, as one died before the MRI; one had the MRI performed elsewhere; and due to poor clinical conditions, one did not undergo the MRI control. The MRI performed 6 months post-SRT showed 2 SD and 4 PD; for five patients, the disease response data at 6 months cannot be reported for the following reasons: four patients died before their MRI (pt 2, 4, 5, 10) (see Table 3) and one patient had the MRI performed elsewhere (pt number 3).

In summary, half of the patients showed a radiological and/or clinical recurrence within 6 months from the end of re-irradiation. 

At tumor progression after SRT, 4 of the 12 patients received second-line chemotherapy with fotemustine, while the remaining ones were referred to palliative care for best supportive care (BSC).

At the time of the completion of the paper’s writing, all patients had died of the disease, with the exception of one whose death occurred due to pneumonia. Median overall survival (OS2) and progression-free survival (PFS2) from the time of re-irradiation were 10.4 months (range: 5.3–19.3) and 5.7 months (range: 1.2–16.1), respectively; 3-, 6- and 12-month OS2 were 92%, 75%, and 42%, respectively; and 3-, 6-, and 12-month PFS2 were 83%, 42%, and 8%, respectively. Patient number 3 died before performing the first follow-up visit, while five patients were proven to be long surviving over the 12 months. 

During follow-up, radionecrosis (RN) was radiologically found in only two patients (pt number 1 and number 6), with an RN rate in our study of 16.6%. Histological confirmation of RN was obtained in one patient, who relapsed after SRT and underwent a second surgery (in this patient, there were concomitant aspects of RN and disease). The median time of brain RN appearance was 6 months. Only one patient showed mild RN-related symptoms (grade 1), such as eye disorders, headache, and fatigue, which required an increase of corticosteroids.

The patient-reported QoL, as assessed by longitudinal collection of the EORTC QLQ-C30, for the 12 patients (100%) who completed both pre- and post-SRT surveys is summarized in Table 3 and Figure 3. 

The median number of surveys per patient was three (range: 2–5 surveys). The questionnaire response rates were 100% (*n* = 12) at T1, 100% (*n* = 12) at T2, 91.7 (*n* = 11) at T3, and finally 50% (*n* = 6) at T4. Compared to scores at T1, at T2 and at T3, the mean global health status score was stable in all patients and there were no significant differences between baseline and 3-month responses across all examined health-related domains, except for a significant negative trend in the EORTC domain of patient-reported functioning in social role QLQ-C30 (*p* < 0.01), and a trend was also recorded for physical functioning (*p* = 0.05). The general quality of life domain did not change after re-irradiation, nor did the cognitive functioning domain (*p* > 0.5).

## 4. Discussion

The management of rGBM remains a daily issue, as therapeutic options are limited and the prognosis remains poor [45]. Different approaches should be considered for recurrent glioblastoma, including systemic agents (chemotherapy, target therapy) or locoregional treatments (radiation therapy and surgery), but their efficacy is still low [41,54,55]. Re-irradiation is a common practice. The adequate selection of patients is a key issue. Age, performance status, target volume, time to progression, type of progression, and site of recurrence are crucial for choosing the technique and fractionation (stereotactic versus hypofractionated versus standard radiotherapy). To date, two prognostic scores have been validated to estimate the prognosis of patients and can be a valid help in the therapeutic choice [56,57,58]. Several reports concerning re-irradiation have been published in the literature; however, the interpretation of clinical data is complex because the current evidence mainly comprises retrospective or institutional case series [44,55,59] with a large variety of schedules and techniques, which makes it difficult to establish a standard approach. 

A large meta-analysis that evaluated 50 studies for a total of 2095 patients with rGBM showed that re-irradiation was associated with 6- and 12-month OS of 73% and 36%, respectively [59], although the majority of studies assessed were retrospective (40/50) and very low quality. One of the studies analyzed in this meta-analysis was conducted at our institute and reported that patients with rGBM showed a median OS after salvage RT of 9.5 months (significantly higher than the patients who were observed), without clinically significant acute morbidity [47]. 

Starting from these promising efficacy and tolerance data, we investigated the possibility of increasing the dose within the target volume while keeping the low dose to the surrounding healthy tissues, with the aim of increasing local control while maintaining a low toxicity profile. The first step was to evaluate the feasibility of a dose painting radiation approach with dose escalation technique guided by ADC values of diffusion-weighted MRI [60]. This dosimetric study proved that ADC-map-guided hypo-fractionated radiotherapy could be carried out and applied to deliver high doses in the GBM recurrent regions. The second step was to evaluate this approach in a prospective phase I study to assess the safety of dose-escalated re-irradiation to relapsed glioblastomas. To our knowledge, the present study is one of the few in the literature with a prospective design, with voxel-based dose escalation, and that evaluates the quality of life in the GBM re-irradiation setting. Table 4 reports the main prospective trials available [61,62,63,64,65,66,67,68,69,70,71,72,73]. 

The toxicity recorded in our study was mild (G1); five patients (42%) showed G2 toxicity (nervous system disorder, headache, fatigue, cognitive disturbance, muscle weakness). In three patients, the acute toxicity was resolved using steroid therapy for three months. The toxicity rate reported in the literature is highly variable: some authors have reported no cases of severe toxicity, while others have indicated a rate of up to 10.5%, including hydrocephalus, dizziness, or worsening of existing neurological symptoms. What is certainly important to point out is that, in our cohort, only one patient (patient 2) experienced severe (G3) toxicity characterized by muscle weakness and fatigue. This patient died 45 days after treatment, and therefore it is not possible to discriminate whether these symptoms were related to the progression that led to death rather than to the treatment-related toxicity. Distinguishing whether the appearance of a symptom is attributable to toxicity related to radiotherapy or disease progression is the main issue. In our study, there was no evidence of a detrimental trend of KPS after re-irradiation, and most of the patients maintained a good quality of life, measured prospectively with validated questionnaires. In our opinion, this is the most important parameter in influencing the prognosis of these patients; indeed, the objective of salvage therapies should be to extend the life expectancy without affecting the residual quality of life. Therefore, patient selection plays a key role in providing the best possible care to each patient, and the risk of re-irradiation-induced toxicity should be estimated considering the previous radiation field, site of recurrence, and target volume. The recurrence volume influences the dose and the radiotherapy technique and therefore plays a prognostic and predictive role of significant importance, as recently shown by Scoccianti et al. [45]. Generally, normo- or hypofractionated re-irradiation schedules for large volumes and SRS and FSRT for small volumes are preferred. For instance, Cho et al. [74] reported a severe toxicity rate of 8% in patients with a median volume of 40 mL, despite of the choice of a schedule with low EQD2 (42.2 Gy). The analysis of the literature suggests that following fractionation schemes based on the target volume should be used in order to achieve a risk of severe effects less than 3.5%, meaning radiosurgery with EQD2 < 65 Gy for a target volume < 12.5 mL, hypofractionated stereotactic radiotherapy with EQD2 < 50 Gy for a target volume up to 35 mL, but conventionally fractionated treatment with EQD2 < 36 Gy for a target volume up to 50 mL [45].

In our series, with the use of voxel-based dose painting, the maximum cumulative dose was less than or equal to 110 Gy, and the EQD2 value, with alpha/beta of 3, was 210 Gy. 

We registered a mean percentage of PTV volume that was included in the 50 Gy isodose of 0.78% (range 0.17–2.26%); consequently, this explains the low toxicity reported and the safety of the treatment. Patient number 2, who showed G3 toxicity, had a PTV of only 33.6 cc. If accepted, this would support the thesis of progression damage rather than acute toxicity. 

The risk of radionecrosis should also be considered. Several studies have reported a variable incidence of radiological diagnosis of radionecrosis (from 4% to 31.1%) [42,45]; Kim et al. reported a very high rate of histologically confirmed radionecrosis up to 12.5% [74]. These same studies did not report whether radionecrosis was associated with neurological symptoms or the required steroids. In our series, radionecrosis was radiologically found in only two patients (16.6%) but histologically confirmed and symptomatic (grade 1) in only one patient, with a median presentation time of 6 months. The one patient who showed mild RN-related symptoms, such as eye disorders, headache, and fatigue, required an increase of corticosteroids. From our experience, the radionecrosis rate with the present re-irradiation technique is low and well manageable with drug therapy, but it is necessary to underline that disease progression and radionecrosis often coexist and can both be responsible for a worsening of symptoms and quality of life. The site of recurrence also assumes critical importance: in our cohort, all the re-irradiated patients had a relapse in the previous treatment field without an increase in the rate of radionecrosis. Moreover, emerging studies evaluating reirradiation in 1–3 sessions associated with new-generation drugs are also of great interest [75].

Although our study was not designed to measure the efficacy of this re-irradiation approach, the median overall survival (OS2) and progression-free survival (PFS2) from the time of re-irradiation were substantially comparable to the literature (10.4 months and 5.7 months, respectively). In the previously reported meta-analysis, the 6-month OS2 and PFS2 were 73% (95% CI 69–77%) and 43% (95% CI 35–50%), respectively; 12-month OS2 and PFS2 were equal to 36% (95% CI 32–40%) and 17% (95% CI 13–20%), respectively [39]. In our series, 3-, 6-, and 12-month OS2 were 92%, 75%, and 42%, respectively, and 3-, 6-, and 12-month PFS2 were 83%, 42%, and 8%, respectively. We obviously cannot provide conclusions on the efficacy of the treatment, but the data seem promising and will certainly need to be confirmed in a future prospective phase II study.

## 5. Conclusions

To date, the treatment of rGBM still represents a medical challenge, and re-irradiation plays a fundamental role in the management of these patients. The modern stereotactic radiation techniques with dose escalation programs can allow the delivery of higher doses to the disease site while keeping the dose low to the surrounding healthy organs, with the aim of increasing local control without increasing toxicity. This approach, investigated in very few studies in the literature, demonstrated a relatively favorable tolerance profile, characterized by moderate acute G1-G2 toxicity and a low rate of radionecrosis. Furthermore, most of the patients did not show a deterioration in the quality in terms of cognitive and social function after re-irradiation, with an overall objective response rate of 30%. The future prospective phase II study will analyze the efficacy in terms of PFS and OS.

## Figures and Tables

**Figure 1 cancers-14-04604-f001:**
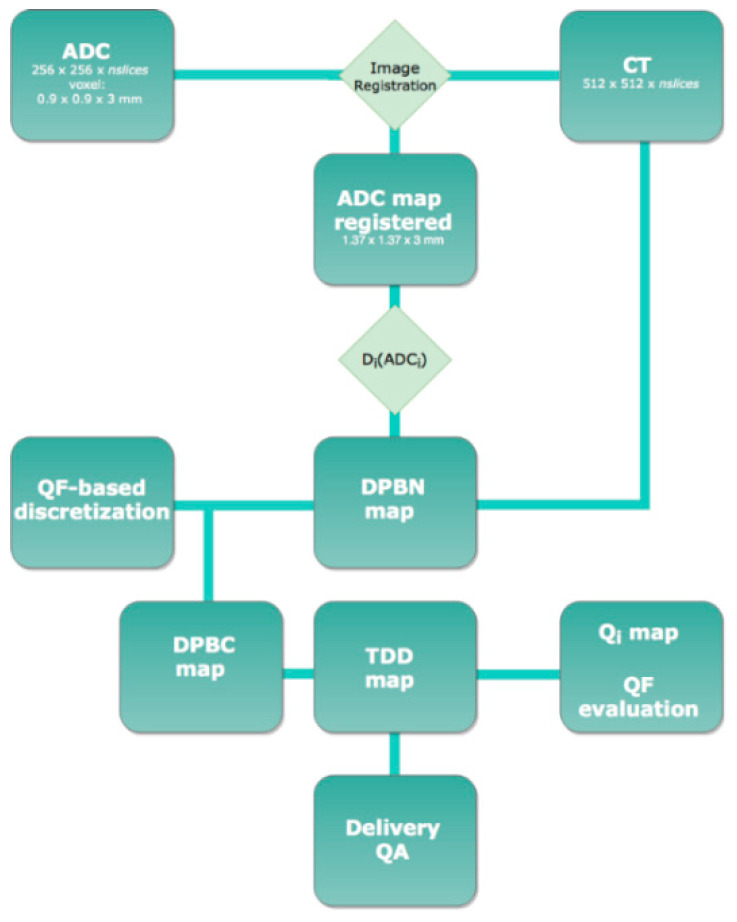
Planning workflow. Workflow of the DPBN planning. The QF-based discretization method was used to identify the target dose levels (DPBC map). Moreover, the treatment planning system (TPS) structured-based dose optimization (TDD map) was conducted. The deliverable optimized dose maps (TDD map) were compared with the DPBN maps and evaluated in terms of QF dose index (QF evaluation). Finally, a delivery quality assurance (Delivery QA) of the plans was carried out and ranked in terms of γ-index.

**Figure 2 cancers-14-04604-f002:**
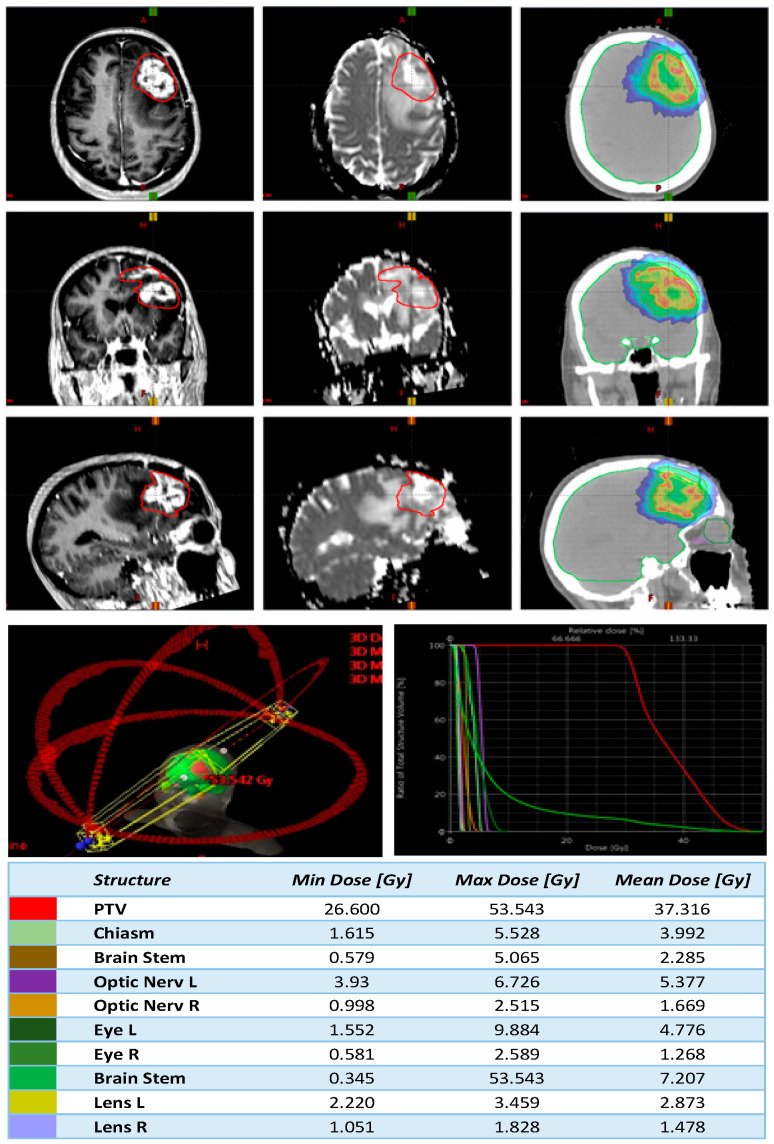
The figure shows an example of T1 W and ADCmap MRI with delineation of GTV, as well as dose distribution. On the bottom, field geometries of the plan, dose volume results, and DWH are reported.

**Figure 3 cancers-14-04604-f003:**
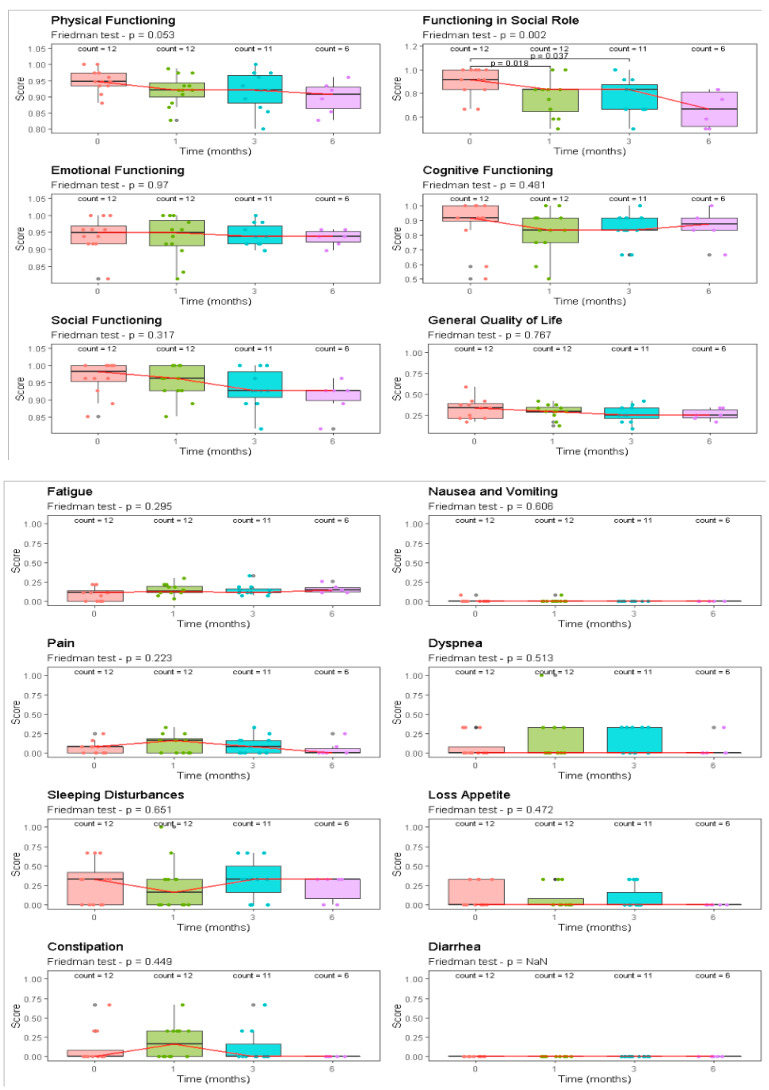
General QoL and social functioning scale summary. Longitudinal comparison of general QoL and social functioning scale scores according to surveillance period. Higher scores represent better satisfaction with patients’ quality of life. p-values through Friedman test and post hoc analysis are shown.

**Table 1 cancers-14-04604-t001:** Main patient and tumor characteristics.

Study Population (*n* = 12)	
Primary Treatment	
TYPE OF SURGERY, *n* (%) – GTR – Partial resection – Biopsy	8 (66.7) 3 (25) 1 (8.3)
HISTOLOGICAL TYPE, *n* (%) – GBM – AA – AO	10 (83.4) 1 (8.3) 1 (8.3)
IDH MUTATION, *n* (%) – WT – Absent – Unknown	4 (33.3) 1 (8.3) 7 (58.4)
MGMT metylation, *n* (%) – Yes – No	7 (58.4) 5 (41.6)
1p19q-codeletion, *n* (%) – Codeleted – No codeleted – Unknown	1 (8.3) 10 (83.4) 1 (8.3)
Radiotherapy dose, *n* (%) – 60 Gy in 30 fr – 45 Gy in 15 fr	11 (91.7) 1 (8.3)
Re-surgery, *n* (%) – Yes – No	2 (16.7) 10 (83.3)
Second-line chemotherapy, *n* (%) – Yes – No	3 (25) 9 (75)
At Recurrence	
GENDER, *n* (%) – Male – Female	7 (58.3) 5 (41.7)
Median age, years (range)	60.5 (51–70)
KPS, *n* (%) – 100 – 90 – 80 – 70	9 (75) 1(8.3) 1 (8.3) 1 (8.3)
GTV, cc – median – range	22.95 5–69.6
PTV, cc – median – range	45.7 15–97.13

Abbreviations: GTR: gross total resection; GBM: glioblastoma; AO: anaplastic oligodendroglioma; AA: anaplastic astrocytoma; IDH: isocitrate dehydrogenase; W-T: wild-type; MGMT: methylguanine-DNA methyl-transferase; Gy: gray; fr: fractions; KPS: Karnofsky Performance Status; GTV: gross tumor volume; PTV: planning target volume.

**Table 2 cancers-14-04604-t002:** Reported acute and late toxicities and outcomes.

Patient	3-m MR I Evaluation	3-m Tox (Grade)	6-m MRI Evaluation	6-m Tox (Grade)	12-m Tox (Grade)	RN	PFS2 (m)	OS2 (m)	Treatment to Progression
1	SD	G1 (headache)	PD	G1 (eye disorders—other, headache, fatigue)	G1 (cognitive disturbance)	Yes	6.9	13.2	Fotemustine
2	NA	G2 (dysphasia), G2 (muscle weakness lower limb), G3 (muscle weakness, fatigue)	NA	NA	NA		1.2	2.8	BSC
3	NA	NA	NA	NA	NA		5.5	11.6	NA
4	PD	G1 (fatigue, muscle weakness lower limb, nausea, eye disorders—other)	NA	NA	NA		2.2	5.3	Fotemustine
5	NA	G1 (ataxia), G2 (nervous system disorders—other, muscle weakness lower limb)	NA	NA	NA		5.3 (date of death)	5.3	BSC
6	SD	G1 (irritability)	SD	G0	G1 (irritability)	Yes	9.3	12.6	BSC
7	PD	G1 (generalized muscle weakness, dysesthesia), G2 (headache)	PD	G1 (ataxia), G2 (nervous system disorders—other)	NA		3.5	6.8	Fotemustine
8	SD	G2 (fatigue, gynecomastia)	SD	G1 (fatigue)	G1 (fatigue)		10.9	19.3	Fotemustine
9	PsP	G1 (headache, fatigue, seizure, conjunctivitis), G2 (eye disorders-other, urinary incontinence, generalized muscle weakness	PD	G2 (seizure, nervous system disorders—other)	NA		2.6	9.2	BSC
10	PD	G1 (dysesthesia, headache)	NA	NA	NA		3.3	6.1	BSC
11	PD	G1 (dysphasia, cognitive disturbance, headache)	PD	G0	G1 (nervous system disorders—other)		4.1	16.5	Fotemustine
12	SD	G1 (fatigue, irritability)	SD	G0	G2 (headache)		11.5	13.5	Surgery

Abbreviations: m: months; MRI: magnetic resonance; RN: radionecrosis; SD: stable disease; PD: progressive disease; NA: not applicable; Tox: toxicity, G: grade; BSC: best supportive care.

**Table 3 cancers-14-04604-t003:** Comparisons of the EORTC QLQ-C30 questionnaire scores at the four time points.

EORTC QLO-C30	Pre Treat		30 Days		90 Days		120 Days		
	(*n* = 12)		(*n* = 12)		(*n* = 12)		(*n* = 12)		*p*
Component	Median [Range]		Median [Range]		Median [Range]		Median [Range]		Friedman
Functional scale									
Physical functioning	95	[88–100]	92	[83–99]	92	[80–100]	91	[83–96]	0.05
Functioning in social role	92	[67–100]	83	[50–100]	83	[50–100]	67	[50–83]	<0.01
Emotional functioning	95	[81–100]	95	[81–100]	94	[90–100]	94	[90–96]	0.97
Cognitive functioning	92	[50–100]	83	[50–100]	83	[67–100]	88	[67–100]	0.48
Social functioning	98	[85–100]	96	[85–100]	93	[81–100]	93	[81–96]	0.32
General quality of life	33	[17–58]	29	[13–42]	25	[8–42]	25	[17–33]	0.77
Symptom scale									
Fatigue	11	[0–22]	13	[4–30]	11	[7–33]	15	[11–26]	0.29
Nausea and vomiting	0	[0–8]	0	[0–8]	0	[0–0]	0	[0–0]	0.61
Pain	8	[0–25]	17	[0–33]	8	[0–33]	0	[0–25]	0.22
Dyspnea	0	[0–33]	0	[0–100]	0	[0–33]	0	[0–33]	0.51
Sleeping disturbances	33	[0–67]	17	[0–100]	33	[0–67]	33	[0–33]	0.65
Loss of appetite	0	[0–33]	0	[0–33]	0	[0–33]	0	[0–0]	0.47
Constipation	0	[0–67]	17	[0–67]	0	[0–67]	0	[0–0]	0.45
Diarrhea	0	[0–0]	0	[0–0]	0	[0–0]	0	[0–0]	-

**Table 4 cancers-14-04604-t004:** Main prospective study regarding glioma re-irradiation.

Author	Number of pts	Re-RT Total Dose (Gy)	No. of Fractions	Median PTV (cc)	Cumulative BED 2 Dose (Gy)	Severe Toxicities	Median OS (m)	PFS (m)
Cabrera et al. (2013) [61]	15	18–25	1–5	n.r.	n.r.	6.3	14.4	3.9
Clarke et al. (2017) [62]	15	27–33	3	n.r.	197.78	n.r.	13	7
Ernst-Stecken et al. (2007) [63]	15	35	5	22.40	157.50	0	12	12
Fields et al. (2012) [64]	10	36	3	54.30	n.r.	n.r.	6	3
Gutin et al. (2009) [65]	25	30	5	34	120	4%	12.5	n.r.
Konget al. (2008) [66]	114	16	1	n.r.	544	0	13	n.r.
Larson et al. (2002) [67]	26	12–20	1	17.2	n.r.	8	9.5	n.r.
Lee et al. (2018) [68]	25	45	15	n.r.	n.r.	32	16	n.r.
Marzano et al. (2011) [69]	22	14–22	1	n.r.	197.28	0	11	n.r.
Møller et al. (2017) [70]	31	29, 5–42	5–10	67	n.r.	9.7		2.8
Schwer et al. (2008) [71]	15	18–36	3	41.3	n.r.	0	10	7
Shi et al. (2016) [72]	17	35	5	n.r.	n.r.	n.r.	n.r.	n.r.
Yoshikawa et al. (2006) [73]	25	13, 9–26, 4	n.r.	19.5	n.r.	n.r.	n.r.	n.r.

Abbreviations: Re-RT: re-irradiation, N: number, PTV: planning tumor volume; OS: overall survival; PFS: progression-free survival; m: months; n.r.: not reported.

## Data Availability

The data presented in this study are available on request from the corresponding author.

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
