# Peer review of "Safety of Inhomogeneous Dose Distribution IMRT for High-Grade Glioma Reirradiation: A Prospective Phase I/II Trial (GLIORAD TRIAL)"

_cancers, 2022, doi:10.3390/cancers14194604_

Round 1

Reviewer 1 Report

The authors present results of a prospective phase I/II study of 12 patients with recurrent GBM treated with a novel technique of functional imaged-guided dose-escalated stereotactic reirradiation using 30 Gy in 5 fractions with a simultaneous integrated boost up to 50 Gy in 5 fractions based on ADC values. No G4 or above toxicity within 6 months occurred, and only 1 grade 3 toxicity occurred. OS2 and PFS were comparable to the literature.

The study is appropriately designed for this novel treatment technique. Prospective data in this space is welcome. The manuscript is overall well-written. I have several major and minor comments as follows.

Major comments:

·         Could the authors clearly and explicitly state the primary endpoint? Was it “acute” toxicity? If so, what was the timeframe to define acute (e.g., 3 months? 6 months?)

·         Throughout the paper, the authors state “no toxicities greater than G4 were recorded.” However, it appears that no G4 toxicity occurred. Should the authors not then state, “no toxicities greater than G3 were recorded” or “no G4 toxicity occurred”?

·         Figure 2 – this figure would greatly benefit from an image of the corresponding ADC map to demonstrate these areas receiving dose escalation.

·         Section 2.4 - Could the authors clarify how target volumes were delineated? Was it FLAIR or enhancement or both which were targeted? Was there a CTV? What was the PTV margin? My understanding is that after this target was contoured, then the dose-escalation based on ADC value was performed within this volume? An example of target volumes with corresponding MRI would be of value.

·         Line 57 and Line 443 – I think that stating this approach demonstrated an “optimal tolerance profile” is an overstatement considering that there was still G1-3 toxicity. A true “optimal” profile would be zero toxicity. Characterizing the toxicity as “relatively favorable” or “tolerable” is more consistent with the presented data. This should be modified in some fashion so that the conclusion is concordant with the data.

Minor comments

·         Line 23 – I suggest specifying that GBM is the most common primary malignant brain tumor, to distinguish it from meningiomas, which are more common.

·         Lines 40-41; 65-66 – Consider adding “with or without tumor treating fields” as part of standard of care, though I realize this may be controversial.

·         Line 127 – “progress” should probably state “having progressed”

·         Figure 1 – the image in the bottom right panel has red underlines from spell-check which should be removed

·         Lines 216-217  - I assume the Dmin and Dmax was defined as 0.03 cc or another volume? Because the values provided contradict those in figure 2. Please clarify.

·         Line 233 is an incomplete sentence

·         Table 1 – “Unknown” is misspelled twice as “Unknow”

·         There is little mention of other recent studies using functional imaging-guided dose-escalation for GBM (for example PMID: 33771703; PMID: 33524546 – though these are in the newly diagnosed setting).

Author Response

Dear reviewer: Thank you for your comments and suggestions. Below you will find the required changes point by point.

Major comments:

  • Could the authors clearly and explicitly state the primary endpoint? Was it “acute” toxicity? If so, what was the timeframe to define acute (e.g., 3 months? 6 months?)

We added in line 141 following sentence: The primary andpoint was the safety, i.e. assessment of acute toxicity reported within 3 months of the end of reirradiation

  • Throughout the paper, the authors state “no toxicities greater than G4 were recorded.” However, it appears that no G4 toxicity occurred. Should the authors not then state, “no toxicities greater than G3 were recorded” or “no G4 toxicity occurred”?

Thanks for the valuable correction. It was a mistake. We made the corrections in the text, abstract, simple summary and results (line 51).

  • Figure 2 – this figure would greatly benefit from an image of the corresponding ADC map to demonstrate these areas receiving dose escalation.

Done.

  • Section 2.4 - Could the authors clarify how target volumes were delineated? Was it FLAIR or enhancement or both which were targeted? Was there a CTV? What was the PTV margin? My understanding is that after this target was contoured, then the dose-escalation based on ADC value was performed within this volume? An example of target volumes with corresponding MRI would be of value.

We added in this section following description: Target volume delineation was performed using ARIA® Oncology Treatment Plan-ning System (TPS). The gross tumor volume (GTV) corresponded to the contrast enhance-ment visible lesion on MRI. Clinical target volume (CTV) was not created for stereotactic radiotherapy planning; therefore, the two volumes GTV and CTV coincided in our series. An isotropic 5 mm expansion was applied to the GTV/CTV to obtain the planning target volume (PTV). The outlined OARs were: brainstraim, lens, eyes, optical chiasma, optical nerves and tormal brain tissue

Moreover . Regarding “in which target the dose escalation is performed”, it is explained in planning paragraph, section 2.5, line 192.

Thanks.

  • Line 57 and Line 443 – I think that stating this approach demonstrated an “optimal tolerance profile” is an overstatement considering that there was still G1-3 toxicity. A true “optimal” profile would be zero toxicity. Characterizing the toxicity as “relatively favorable” or “tolerable” is more consistent with the presented data. This should be modified in some fashion so that the conclusion is concordant with the data.

You are right. we changed "optimal" to "acceptable" or "relatively favorable" in abstract and main text.

Thank you very much

Minor comments

  • Line 23 – I suggest specifying that GBM is the most common primary malignantbrain tumor, to distinguish it from meningiomas, which are more common.

Done. Thank you again

  • Lines 40-41; 65-66 – Consider adding “with or without tumor treating fields” as part of standard of care, though I realize this may be controversial.

The addition of this sentence could create misunderstandings, we would prefer not to add it.

  • Line 127 – “progress” should probably state “having progressed”

Thank you. We corrected it.

  • Figure 1 – the image in the bottom right panel has red underlines from spell-check which should be removed

We modified it.

  • Lines 216-217  - I assume the Dmin and Dmax was defined as 0.03 cc or another volume? Because the values provided contradict those in figure 2. Please clarify.

We modified the line as follow: establishing 1 cc the maximum volume that receives a dose greater than 50 Gy. Thanks.

  • Line 233 is an incomplete sentence

We edited the sentence. Thanks

  • Table 1 – “Unknown” is misspelled twice as “Unknow”

We edited the table. Thanks

  • There is little mention of other recent studies using functional imaging-guided dose-escalation for GBM (for example PMID: 33771703; PMID: 33524546 – though these are in the newly diagnosed setting).

kind reviewer. You are right, some insights are missing from the text, including the one you cited. However, given the large amount of information that we had to concentrate in a single article, it was not possible to deepen everything. A new article is already being written which will aim at a broader analysis of the technique in which the dose escalation will be analyzed also in the treatment of newly diagnoses and we will add also these articles. Thank you.

Reviewer 2 Report

This is a well written and nice presentation of a small phase 1 prospective study of a dose-painting by numbers approach to heterogeneous dose delivery based upon ADC mapping for re-irradiation of GBM. I have the following comments:

1) It would be helpful to see more dosimetric information. Since the main premise of this work is around the possible advantages of a ADC map based dose painting technique, it would be helpful to know how much dose escalation was achieved for patients, and the degree of heterogeneity in plans. Patterns of failure analysis in relation to voxel risk and dose would be interesting as well, but I assume the authors are saving this for future work or additional report. 

Author Response

Dear reviewer: Thank you for your comments and suggestions. Below you will find the required changes point by point.

 This is a well written and nice presentation of a small phase 1 prospective study of a dose-painting by numbers approach to heterogeneous dose delivery based upon ADC mapping for re-irradiation of GBM. I have the following comments:

Dear Reviewer, Thanks for appreciating the work

  • It would be helpful to see more dosimetric information. Since the main premise of this work is around the possible advantages of a ADC map based dose painting technique, it would be helpful to know how much dose escalation was achieved for patients, and the degree of heterogeneity in plans. Patterns of failure analysis in relation to voxel risk and dose would be interesting as well, but I assume the authors are saving this for future work or additional report. 

This study lends itself to various clinical and technical talking points. Just as you guessed, we are writing a further article just to respond to your observations, as it was not conceivable to adequately summarize everything in a single work.

Thank you so much.

Reviewer 3 Report

The re-irradiation in an "hot topic" in field of high grade glioma recurrence, in which SRT is an emerging treatment approach. Moreover, the present study is an interesting example of radiotherapy guided by functional imaging.

Overall, the research is well structured, the methodology is adequate, the results are clear reported and discussed. 

I suggest to add in the discussion the possible benefit by the association between SRT and Regorafenib [see Gregucci F, et al. Radiosurgery and Stereotactic Brain Radiotherapy with Systemic Therapy in Recurrent High-Grade Gliomas: Is It Feasible? Therapeutic Strategies in Recurrent High-Grade Gliomas. J Pers Med. 2022 Aug 20;12(8):1336. doi: 10.3390/jpm12081336].

Author Response

Dear reviewer: Thank you for your comments and suggestions. Below you will find the required changes point by point.

The re-irradiation in an "hot topic" in field of high grade glioma recurrence, in which SRT is an emerging treatment approach. Moreover, the present study is an interesting example of radiotherapy guided by functional imaging.

Overall, the research is well structured, the methodology is adequate, the results are clear reported and discussed.

Dear Reviewer, Thanks for appreciating the work

I suggest to add in the discussion the possible benefit by the association between SRT and Regorafenib [see Gregucci F, et al. Radiosurgery and Stereotactic Brain Radiotherapy with Systemic Therapy in Recurrent High-Grade Gliomas: Is It Feasible? Therapeutic Strategies in Recurrent High-Grade Gliomas. J Pers Med. 2022 Aug 20;12(8):1336. doi: 10.3390/jpm12081336].

Thanks for the welcome suggestion. We have proceeded to insert the article.

Reviewer 4 Report

The manuscript can be accepted after the authors correct the following comments:

1.     The title is too long, unclearly and insufficiently reflects its content. Moreover, should be rephrased with good academic writing.

2.     Please note that the simple summary section should be moved to the last paragraph at end of the manuscript (before conclusion section).

  1. Please note that the introduction part should divide to at least four comprehensive paragraphs.

  1. Add a new paragraph at the introduction section about radiotherapy with modern references.

5.     Please make double check about the academic writing (needs native speakers in English).

6.     Lines 120-122 in the method section should be deleted.

7.     The figures resolution is not clear. Please increase the resolution.

  1. The conclusion section should be reflecting the results in a good way.

  1. Make sure that all sentences are linked together.

  1. If the experiments are done in the hospital, so kindly insert the IRB approval and code.

Author Response

Dear reviewer: Thank you for your comments and suggestions. Below you will find the required changes point by point.

  1. The title is too long, unclearly and insufficiently reflects its content. Moreover, should be rephrased with good academic writing.

We modified the title as follow: Safety of inhomogeneous dose distribution IMRT for high-grade gliomas reirradiation: a prospective phase I/II trial (GLIORAD TRIAL).

  1. Please note that the simple summary section should be moved to the last paragraph at end of the manuscript (before conclusion section).

Done, thanks

  1. Please note that the introduction part should divide to at least four comprehensive paragraphs

Done, Thanks.

  1. Add a new paragraph at the introduction section about radiotherapy with modern references

Dear reviewer. We appreciated your comment. However, in our opinion, the introduction already appears very rich. We also believe that today radiotherapy is an extremely well-known technique and the addition of a further paragraph would make the text burdensome.

  1. Please make double check about the academic writing (needs native speakers in English).

The text was proofread by a native English speaker prior to submission to the journal. However, we have proceeded to modify the reported parts

  1. Lines 120-122 in the method section should be deleted

Done, Thanks.

  1. The figures resolution is not clear. Please increase the resolution

Modified , thanks

  1. The conclusion section should be reflecting the results in a good way.

We edited the conclusions.

  1. Make sure that all sentences are linked together.

We  edited it as required by rev. 1 too.

  1. If the experiments are done in the hospital, so kindly insert the IRB approval and code.

Done.